# Potential Nephroprotective Effect of uPA against Ischemia/Reperfusion-Induced Acute Kidney Injury in αMUPA Mice and HEK-293 Cells

**DOI:** 10.3390/biomedicines12102323

**Published:** 2024-10-12

**Authors:** Heba Abd Alkhaleq, Israel Hacker, Tony Karram, Shadi Hamoud, Aviva Kabala, Zaid Abassi

**Affiliations:** 1Department of Physiology and Biophysics, Rappaport Faculty of Medicine, Technion-Israel Institute of Technology, Haifa 31096, Israel; heba.abd.alkhaleq@gmail.com (H.A.A.); yisraelhacker@gmail.com (I.H.); avivak@g.technion.ac.il (A.K.); 2Department of Vascular Surgery, Rambam Health Care Campus, Haifa 3109601, Israel; t_karram@rambam.health.gov.il; 3Internal Medicine E, Rambam Health Care Campus, Haifa 3109601, Israel; s_hamoud@rambam.health.gov.il; 4Laboratory Medicine, Rambam Health Care Campus, Haifa 3109601, Israel

**Keywords:** acute kidney injury, urokinase plasminogen activator, inflammation, fibrosis, ACE-2, eNOS

## Abstract

**Background/Objectives:** The incidence of acute kidney injury (AKI) has been steadily increasing. Despite its high prevalence, there is no pathogenetically rational therapy for AKI. This deficiency stems from the poor understanding of the pathogenesis of AKI. Renal ischemia/hypoxia is one of the leading causes of clinical AKI. This study investigates whether αMUPA mice, overexpressing the urokinase plasminogen activator (uPA) gene are protected against ischemic AKI, thus unraveling a potential renal damage treatment target. **Methods:** We utilized an in vivo model of I/R-induced AKI in αMUPA mice and in vitro experiments of uPA-treated HEK-293 cells. We evaluated renal injury markers, histological changes, mRNA expression of inflammatory, apoptotic, and autophagy markers, as compared with wild-type animals. **Results:** the αMUPA mice exhibited less renal injury post-AKI, as was evident by lower SCr, BUN, and renal NGAL and KIM-1 along attenuated adverse histological alterations. Notably, the αMUPA mice exhibited decreased levels pro-inflammatory, fibrotic, apoptotic, and autophagy markers like TGF-β, IL-6, STAT3, IKB, MAPK, Caspase-3, and LC3. By contrast, ACE-2, p-eNOS, and PGC1α were higher in the kidneys of the αMUPA mice. In vitro results of the uPA-treated HEK-293 cells mirrored the in vivo findings. **Conclusions:** These results indicate that uPA modulates key pathways involved in AKI, offering potential therapeutic targets for mitigating renal damage.

## 1. Introduction

Acute kidney injury (AKI) is characterized by abrupt renal damage that results in a decline in kidney function within hours, days, or weeks. Renal ischemia and hypoxia are among the leading causes of clinical AKI [1,2,3], where the incidence of AKI steadily increasing over time [4]. Despite the high prevalence of AKI and its association with an alarming increase in morbidity and mortality, therapeutic approaches remain disappointing, relying mainly on supportive measures [1,2,3]. The limited advancements can be attributed to a minimal understanding of the pathogenesis of AKI. Since AKI poses an important clinical problem [3,5,6], current criteria for diagnosing and classifying acute kidney dysfunction typically rely on measuring urine flow and, mainly, serum creatinine (SCr). Relying on SCr levels for diagnosis of AKI is majorly limiting, as this test has considerable restrictions [7]: SCr takes time to re-adjust in response to changes in the glomerular filtration rate (GFR), and subtle transient changes in the GFR may remain concealed. Additionally, it cannot differentiate altered glomerular hemodynamics and pre-renal failure from true renal tissue injury, necessitating additional clinical and laboratory diagnostic tools [8]. During recent decades, great development in molecular biology has led to major advances in the search for novel biomarkers for early detection of AKI [9], as evident by the discovery of several key biomarkers with various specificity and sensitivity, including neutrophil gelatinase-associated lipocalin (NGAL) and kidney injury molecule-1 (KIM-1) [9,10,11,12]. NGAL, a sensitive biomarker for renal tubular injury, is one of the most prominently upregulated genes in the kidney after AKI, particularly in distal nephron segments [9,11]. Leukocytes, the loop of Henle, and collecting ducts all express NGAL [13]. NGAL is also present in the proximal tubules [14]. KIM-1 is a protein highly expressed in tubular epithelial cells during renal dysfunction. It is sustained at high levels and released into both urine and blood, with greater release in urine, directly promoting the removal and regeneration of necrotic tubular cells [15]. KIM-1 is highly expressed in the apical membrane of proximal tubule cells following ischemia and nephrotoxic damage [16].

To investigate the mechanisms underlying AKI, αMUPA transgenic mice were utilized as a model to study the potential involvement of urokinase plasminogen activator (uPA), a serine protease known for its roles in tissue remodeling and cell migration, including fibrinolysis and in kidney pathophysiology. Notably, αMUPA transgenic mice overexpress uPA primarily in their brain [17]. Several phenotypic changes observed in αMUPA mice include spontaneous reduction of food intake, growth and body weight retardation, and increased longevity [17]. Notably, long-lived transgenic αMUPA mice exhibit a protective effect on the heart against functional and inflammatory damage following myocardial infarction (MI) [18,19] and sepsis induced by LPS treatment [20]. Furthermore, we have demonstrated that αMUPA mice exhibit nephroprotection against I/R-induced AKI [21,22]. We postulate that the mechanisms underlying the nephroprotection involve the indirect effect of uPA through upregulation of angiotensin-converting enzyme-2 (ACE-2) and endothelial nitric oxide synthase (eNOS). The ACE-2 counteracts the action of the ACE by converting Ang II to Ang 1-7. Ang 1-7 then binds to the Mas receptor (MasR), in various target organs [23], neutralizing the effects of the ACE–Ang II–AT1 axis [23]. The ACE2–Ang 1-7–Mas axis increases NO and prostaglandins concentrations which promote vasodilation, antiproliferative activity, and diuresis along with decreasing oxidative stress. These processes probably protect the kidneys from ischemic and non-ischemic damage [24,25,26,27]. Notably, the ACE-2 is expressed in the proximal tubules more robustly than in the glomeruli [28]. In this context, downregulation of the ACE-2 in the kidney has been linked to various renal disorders [28,29]. Mice subjected to I/R-AKI displayed a decreased tubular ACE-2 in their kidneys, accompanied by elevated tubular NGAL staining [30]. Upregulation of the ACE-2 pathway results in the activation of the peroxisome proliferator-activated receptor-gamma coactivator (PGC1α) [31], an important modulator of mitochondrial functions to combat oxidative stress, inflammation, and fibrosis [32]. NO is created by eNOS, an isoform that is predominantly present in renal arteries and glomeruli [33,34]. Previous studies have demonstrated that NO plays a role in controlling renal tubular function by inhibiting the reabsorption of salt and water, consequently increasing natriuretic and diuresis [35]. The vasoconstrictive effect of angiotensin II is strongly modulated by NO in preglomerular afferent arterioles [36]. In addition to these indirect effects, we assume that the uPA has direct renal beneficial effects, including on tubular epithelial cells [37]. Specifically, one study showed that the uPA prevents renal stone formation [38], while another study demonstrated that mice deficient in uPA were vulnerable to severe renal dysfunction. Additionally, this study showed that plasminogen and plasminogen activators have a protective effect in acute inflammatory renal damage associated with crescentic glomerulonephritis [39].

In the current study, we applied both in vivo and in vitro protocols to explore the nephroprotective effects of the uPA. Specifically, we tested the impact of I/R-AKI on kidney function and damage in the αMUPA mice compared with the WT animals. In addition, we examined the direct effect of the uPA on cultured HEK-293 cells (proximal tubule cells of the kidney) and HEK cells overexpressing the ACE-2 by evaluating the mRNA expression and protein abundance of genes that were recognized to play a nephroprotective role in the in vivo protocol.

## 2. Materials and Methods

### 2.1. Animals

Wild-type FVB/N and genetically modified αMUPA mice were sourced from Prof. Ruth Miskin at the Weizmann Institute of Science (Rehovot, Israel) and were bred in the Rappaport Faculty of Medicine (Haifa, Israel). These mice were kept in groups of five per group in a temperature-controlled environment and provided with a standard mouse diet. All experiments were performed following the regulations of the Committee for the Supervision of Animal Experiments at the Technion (IIT IL 097-06-19, IL-191-12-23).

Induction of AKI: Both the WT and αMUPA mice, aged 12 weeks (*n* = 6–13 per subgroup), underwent anesthesia using 100 mg/kg ketamine and 10 mg/kg xylazine. The animals were positioned on their backs on a temperature-regulated surface at 37 °C. A surgical incision was made to access the abdominal cavity, and their intestines were kept moist with saline-dampened gauze to prevent dehydration. In the experimental groups, the renal arteries were clamped, using vascular clamps, for 30 min to induce ischemia. Blood flow return was confirmed visually once the clamps were removed. The abdominal incisions were then sutured, and the mice were placed back in their housing for recovery. Reperfusion time was calculated starting from the clamp removal, and its impact on serum creatinine (SCr) and blood urea nitrogen (BUN) levels, histological sections, and biomarker expressions were analyzed at the 24-h mark. Mice that underwent sham operations without renal artery clamping were used as the control group.

### 2.2. Blood Variable and Renal Function Assessment

At 24 h post-AKI induction, the animals were re-anesthetized with 100 mg/kg ketamine and 10 mg/kg xylazine. 500 µL blood samples were drawn from the left ventricle of the heart and centrifuged at 1500 rpm for 10 min at 4 °C. SCr and BUN serum concentrations were measured using commercial kits (Siemens, Berlin, Germany) on a Dimension RXL auto-analyzer (Siemens, Germany).

### 2.3. Histopathology

Kidneys from the different experimental groups were harvested, fixed in 4% formalin (Merck, Herzliya, Israel), and embedded in paraffin blocks. Sections were sliced and stained with hematoxylin and eosin (H&E). For the paraffin-embedded sections, tissues were rehydrated by standard methods, which involved xylene (Merck, Herzliya, Israel) treatments (three times) and rehydration using graded alcohols (100% and 95%) followed by rinsing with distilled water. Paraffin sections were then stained with hematoxylin (Merck, Herzliya, Israel) for two minutes, washed with tap water, and allowed to dry. Subsequently, eosin (Merck, Herzliya, Israel) staining was applied for two minutes, followed by the usual dehydration steps, including a series of alcohol and xylene washes. The slides were subsequently sealed using a DPX (Merck, Herzliya, Israel) slide-mounting medium. A histological assessment of the kidney tissue was conducted, analyzing the presence of casts, necrosis, and inflammation in both the WT and αMUPA mice that underwent sham surgery or AKI induction.

### 2.4. Real-Time PCR

Total RNA was purified from frozen mice kidney samples (cortex and medulla) and from the frozen HEK-293 and ACE-2 overexpressing cells using TRIzol reagent (Thermo Fisher Scientific, Waltham, MA, USA). Subsequently, cDNA was synthesized from the purified RNA following the manufacturer’s protocol using the Maximal first strand cDNA synthesis kit (Bioconsult, Jerusalem, Israel) for RT-qPCR. Quantitative real-time PCR analysis was then conducted using PerfeCTa SYBR Green (Quantabio, Beverly, MA, USA) with specific primers for the target genes, and the data were analyzed using the 7500 Real Time PCR System (Applied Biosystems, RHENIUM 8440, Foster City, CA, USA). The mRNA levels of the various genes (uPA, uPAR, eNOS, ACE-2, IL-6, Leptin, LC3, KIM-1, NGAL and PGC1α) were standardized to mRNA levels of Rpl13a in mice, referred to as the housekeeping gene, and GAPDH in cultured cells. Relative to the normalized values obtained for the WT mice at baseline (in vivo) and for the HEK-293 cells without treatment (in vitro), fold change was measured.


Mouse primers:


uPA: -F-AGAGTCTGAAAGTGACTATCTC,-R-CCTTCGATGTTACAGATAAGC

uPAR: -F-TCTGGATCTTCAGAGCTTTC,-R-GCCTCTTACGGTATAACTCC

PAI-1: -F-AGCAACAAGTTCAACTACAC,-R-CTTCCATTGTCTGATGAGTTC

InsR: -F-AAGACCTTGGTTACCTTCTC,-R-GGATTAGTGGCATCTGTTTG

eNOS: -F-AAAGCTGCAGGTATTTGATG,-R-AGATTGCCTCTATTTGTTGC

ACE-2: -F-CATTTGCTTGGTGATATGTG,-R-GCCTCTTGAAATATCCTTTCTG

MasR: -F-GTTTAAGGAACTCTGGAAGATG,-R-TTAGTCAGTTAGTCAGTGGC

Renin: -F-AGCCAAGGAGAAGAGAATAG,-R-CTCCTGTTGGGATACTGTAG

IL-6: -F-GTCTATACCACTTCACAAGTC,-R-TGCATCATCGTTGTTCATAC

TLR4: -F-TCCCTGCATAGAGGTAGTTCC,-R-TCCAGCCACTGAAGTTCTGA

Leptin: -F-ACATTTCACACACGCAGTCGG,-R-GGACCTGTTGATAGACTGCCA

LC3: -F-GAACCGCAGACGCATCTCT,-R-TGATCACCGGGATCTTACTGG

P62: -F-AATGTGATCTGTGATGGTTG,-R-GAGAGAAGCTATCAGAGAGG

Galectin-8: -F-ATATACAAAAGCCAGGCAAG,-R-CAAATGCTTTCACATTGAGG

TGF-β: -F-GGATACCAACTATTGCTTCAG,-R-TGTCCAGGCTCCAAATATAG

Caspase-3: -F-CATAAGAGCACTGGAATGTC,-R-GCTCCTTTTGCTATGATCTTC

Caspase-7: -F-CAAAACCCTGTTAGAGAAACC,-R-CCATGAGTAATAACCTGGAAC

KIM-1: -F-CTGGAGTAATCACACTGAAG,-R-AAGTATGTACCTGGTGATAGC

NGAL: -F-ATATGCACAGGTATCCTCAG,-R-GAAACGTTCCTTCAGTTCAG

PGC1α: -F-TCCTCTTCAAGATCCTGTTAC,-R-CACATACAAGGGAGAATTGC 

Rpl13a: -F-AAGCAGGTACTTCTGGGCCG,-R-GGGGTTGGTATTCATCCGCT


Human primers:


uPA: -F-GAAAACCTCATCCTACACAAG,-R-ATTCTCTTTTCCAAAGCCAG

uPAR: -F-AATCCTGGAGCTTGAAAATC,-R-CAGTCAATGAGGAAAGTCTC

PAI-1: -F-ATCCACAGCTGTCATAGTC,-R-CACTTGGCCCATGAAAAG

InsR: -F-GATCCAATCTCAGTGTCTAAC,-R-CCTTTGAGGCAATAATCCAG

eNOS: -F-CAACCCCAAGACCTACG,-R-CGCAGACAAACATGTGG

ACE-2: -F-TCATTATGAGGACTATGGGG,-R-CTCTTCAAAGGTATGTTCCAC 

MasR: -F-ATCCCTTCACTGTCTACATC,-R-TAGCCAAACAGAAAAGTCAC

Renin: -F-TTATGTCGTGAAGTGTAACG,-R-TGCACAGCTTTTTACTACTG 

IL-6: -F-GCAGAAAAAGGCAAAGAATC,-R-CTACATTTGCCGAAGAGC 

TLR4: -F-GATTTATCCAGGTGTGAAATCC,-R-TATTAAGGTAGAGAGGTGGC

Leptin: -F-TCAATGACATTTCACACACG,-R-TCCATCTTGGATAAGGTCAG 

LC3: -F-ATAGAACGATACAAGGGTGAG,-R-CTGTAAGCGCCTTCTAATTATC

P62: -F-TGTGAATTTCCTGAAGAACG,-R-TCGATATCAACTTCAATGCC 

Galectin-8: -F-AAAATGTTCCAAAGTCTGGC,-R-TGACACATAGTTCATAGGTGG

TGF-β: -F-TGTACCAGAAATACAGCAAC,-R-CAAAAGATAACCACTCTGGC

Caspase-3: -F-AAAGCACTGGAATGACATC,-R-CGCATCAATTCCACAATTTC

Caspase-7: -F-AAGCCATGGAGAAGAAAATG,-R-CCTGAATGAAGAAGAGTTTGG 

KIM-1: -F-CTTACACAACAGATGGGAATG,-R-CCAGCATAGATTCCTTTAGTG

NGAL: -F-GGAAAAAGAAGTGTGACTACTG,-R-GTAACTCTTAATGTTGCCCAG 

PGC1α: -F-GCAGACCTAGATTCAAACTC,-R-CATCCCTCTGTCATCCTC 

GAPDH: -F-TCGGAGTCAACGGATTTG,-R-CAACAATATCCACTTTACCAGAG

### 2.5. Western Blot

Samples of kidney tissue (20 mg), including cortex and medulla (in vivo), were homogenized in a lysis buffer, while cultured cells (in vitro) were mixed with a lysis buffer. Protein quantification was performed using a Bradford (Merck, Herzliya, Israel) commercial assay. Protein samples (50 μg) were then electrophoresed on a sodium dodecyl sulfate (SDS) polyacrylamide gel (10%) under denaturing conditions and electro-transferred to nitrocellulose membranes for 1.5 h at 100 V. Membranes were blocked with 5 percent BSA in Tris-buffered saline (TBS) for 1 h at room temperature. Primary antibodies were diluted in TBST at concentrations ranging from 1:200 to 1:1000 with 5% BSA (Merck, Herzliya, Israel) and incubated overnight at 4 °C. To serve as an internal control, immuno-detection of GAPDH with monoclonal anti-GAPDH antibodies was conducted. HRP-conjugated secondary antibodies were applied for 45 min at room temperature at a concentration of 1:15,000. The signal was detected using enhanced chemiluminescence (ECL) substrate, and images were captured with an ImageQuant LAS 4000 system.

### 2.6. Cell Culture

HEK-293 cells (both normal and ACE-2 overexpressing) were cultured in DMEM (Merck, Herzliya, Israel) high glucose with 4.5 g/L D-glucose and 4 mM L-alanine-L-glutamine containing 10% fetal bovine serum (FBS) (Merck, Herzliya, Israel). The cultures were maintained in a humidified atmosphere of 5% CO2 and 95% air at 37 °C. Experiments were conducted 2 days following incubation.

uPA treatment: uPA (5000 units/mL, TauroLock, TauroPharm GmbH, Waldbüttelbrunn, Germany) was diluted in FBS-free medium and applied to cells at a concentration of 10 µL/mL. The cells were then maintained for 24 h. At the end of the treatment period, cells were collected and stored at −80 °C for Western blot and qPCR analysis of the relevant genes.

### 2.7. Statistical Analysis

Animals were randomly assigned to the experimental group. The results are presented as means ± SEM. Statistical significance was tested for comparisons between the WT and αMUPA mice (in vivo) and between the HEK-293 cells with/out uPA and ACE-2 overexpressing cells (in vitro) using unpaired Student’s t-tests. We applied Prism 10 for statistical analysis. A *p*-value of <0.05 was considered statistically significant.

## 3. Results

### 3.1. Kidney Function and Renal Injury Biomarkers

Induction of AKI increased the SCr measured 24 h after renal injury in both the WT and αMUPA mice; however, the latter subgroup exhibited a significantly attenuated elevation in SCr compared to the WT mice (Figure 1A). Similarly, the elevation of BUN following AKI was less pronounced in the αMUPA mice compared to the WT mice. (Figure 1B). Noteworthy, both the WT and αMUPA mice displayed significant increases in renal expression of NGAL and KIM-1, biological markers of AKI, following AKI (Figure 1C,D). The WT mice were more susceptible to AKI, as evidenced by a profound elevation in NGAL expression ~100-fold (Figure 1C), compared to just a ~35-fold increase in the αMUPA mice. The attenuated I/R injury 24 h following AKI in the αMUPA mice demonstrates that these mice exhibit increased tolerance to ischemic renal stress. The addition of uPA to the cultured normal HEK-293 cells and the ACE-2 overexpressing cells resulted in a decreased NGAL expression in the HEK-293 (Figure 1E), but not in the ACE-2 overexpressing cells. The latter exhibited a reduced NGAL expression at baseline, which was unaffected by the addition of the uPA treatment (Figure 1E). Treatment with uPA did not affect the expression of KIM-1 in either the normal HEK-293 cells or the ACE-2 overexpressing cells (Figure 1F).

### 3.2. Renal Histology

Figure 2 depicts the renal histological alterations in the WT and αMUPA mice subjected to I/R-AKI. Distinct patterns in the histological renal response were observed in the WT and αMUPA kidneys following AKI, as evidenced visually and microscopically, suggesting a protective role of the uPA against renal ischemia (Figure 2). The WT mice exhibited more severe kidney tissue injury, including tubular collapse, loss of the brush border, and cellular detachment from tubular basement membranes (Figure 2). Additionally, necrosis and the presence of hyaline casts were observed in the outer medulla, whereas congestion was more intense in the inner medulla. By contrast, the αMUPA mice exhibited attenuated kidney injury in response to I/R-AKI.

### 3.3. Renal uPA and uPA Receptor

The WT mice exhibited a reduction in renal uPA expression/abundance (Figure 3A,D) along with a significant elevation in uPA receptor expression (~13 fold) following AKI (Figure 3B). The αMUPA mice displayed an enhanced uPA receptor mRNA expression following AKI, but to a lesser extent than the WT mice (Figure 3B). A specific glycoprotein, plasminogen activator inhibitor 1 (PAI-1), binds to the two-chain of uPA and blocks the proteolytic action of the uPAR-bound uPA [40]. Analysis of the renal PAI-1 expression revealed that the PAI-1 expression was increased in the WT mice following AKI (~80 fold) (Figure 3C). By contrast, the αMUPA mice did not display such significant changes in the expression of renal PAI-1 compared to the remarkable enhancement in PAI-1 following AKI in the WT mice (Figure 3C). The in vitro study revealed that the uPA treatment resulted in an increase in uPA mRNA expression and abundance in the HEK-293 cells (Figure 3F,I) but decreased this parameter in the ACE-2 overexpressing cells (Figure 3F,I). The uPAR expression increased in the HEK-293 following the uPA treatment (Figure 3G), but uPAR immunoreactivity decreased significantly in both the HEK-293 and ACE-2 overexpressing cells following 24 h of treatment with uPA (Figure 3J). Additionally, the uPA treatment led to a decline in the expression of PAI-1 in both cultured cell types (Figure 3J).

### 3.4. Renal Leptin, Insulin Receptor and PGC1α

Figure 4 illustrates the expression of three major regulators of kidney function across the various experimental groups. The insulin receptor (InsR) plays a key role in maintaining both glomerular and tubular function [41]. Reduced insulin signaling, such as that found in insulin-resistant conditions, can lead to significant renal issues, including albuminuria, glomerular disease, and hypertension [41]. Following AKI, both the WT and αMUPA mice demonstrated a notable decrease in renal InsR expression. However, the decline was more pronounced in the WT mice compared to αMUPA mice (Figure 4A). Additionally, PGC-1α expression in the kidneys of both the WT and αMUPA mice was lower following AKI (Figure 4B). Given the beneficial renal effects of PGC-1α, we assume that the observed over expression of PGC-1α in the αMUPA mice following AKI may be involved in the attenuated susceptibility of these mice to ischemic injury associated with AKI. With regards to renal leptin expression following AKI, the WT mice displayed a significant elevation (Figure 4C) while the αMUPA mice did not demonstrate any change in renal leptin expression following AKI.

The in vitro protocol revealed that the uPA treatment did not affect InsR mRNA expression in both the HEK-293 cells and ACE-2 overexpressing cells compared with cultured cells without treatment (Figure 4D). Following the uPA treatment, PGC-1α expression increased in the HEK-293 cells (Figure 4E); however, it did not change in the ACE-2 overexpressing cells (Figure 4E). Of notice, the HEK-293 and ACE-2 overexpressing cells did not express leptin.

### 3.5. Renal Expression of Inflammatory and Fibrotic Markers

The WT and αMUPA mice showed a significant increase in renal Interleukin 6 (IL-6), a proinflammatory cytokine, expression 24 h following AKI (Figure 5A). Importantly, the WT mice displayed a more pronounced elevation in IL-6 than the αMUPA animals (Figure 5A). Toll like receptor 4 (TLR-4) was unchanged in the experimental groups 24 h post-AKI (Figure 5B). Analysis of renal immunoreactive levels of pro-inflammatory markers: phosphorylated signal transducer and activator of transcription 3 (p-STAT3), Cathepsin L, I Kappa B (IKB), and mitogen-activated protein kinase (MAPK) showed that the WT mice exhibit an increased elevation in renal expression of these markers following AKI. Conversely, the αMUPA mice displayed a reduced elevation of these proinflammatory markers (Figure 5E–H).

AKI induction provoked renal transforming growth factor β (TGFβ) expression in both the WT and αMUPA mice (Figure 5C). However, the increase obtained in the WT mice was more pronounced than the observed rise in the αMUPA mice subjected to AKI. Of interest, the αMUPA mice did not show alterations in renal TGFβ immunoreactive levels following AKI induction (Figure 5I), where the WT mice showed high levels of TGFβ immunoreactivity compared with the αMUPA mice.

The addition of uPA to both the HEK-293 and ACE-2 overexpressing cultured cells did not affect IL-6 mRNA expression in both cell types. It is worth noting that the ACE-2 overexpressing cells had both a decreased basal IL-6 expression as well as after the uPA treatment when compared with the HEK-293 cells (Figure 5J). IL-6 immunoreactivity demonstrated that the uPA treatment significantly decreased IL-6 levels in both the HEK-293 and ACE-2 overexpressing cells (Figure 5N). An analysis of renal immunoreactive levels of the pro-inflammatory markers STAT-3, p-STAT3, IKB, and MAPK post-uPA treatment showed reduced levels of these immunoreactive proteins in both the HEK-293 cells and ACE-2 overexpressing cells (Figure 5L,M,O,P). Moreover, the uPA treatment led to a significant reduction in the expression and abundance of the fibrotic marker TGF-β in both the HEK-293 cells and the ACE-2 overexpressing cells (Figure 5K,Q).

These results demonstrate that the uPA plays a beneficial role in the observed nephroprotection against I/R-induced AKI by ameliorating inflammatory and fibrotic processes in αMUPA mice and cultured cells.

### 3.6. Renal Apoptotic and Autophagy Markers: Renal Apoptotic and Autophagy Markers

Renal expression of Caspase-3 increased following AKI in the WT mice, but decreased significantly in the αMUPA animals, as presented in Figure 6A. Conversely, kidney expression of Caspase-7 did not change in the WT and αMUPA mice following AKI (Figure 6B). The renal expression of various autophagy markers, including the microtubule-associated protein 1A/1B-light chain 3 (LC3), P62 and Galectin-8 were markedly reduced in the αMUPA mice following AKI (Figure 6C,D,F). LC3 immunoreactivity increased following AKI in the WT mice, but did not change in the αMUPA mice (Figure 6E).

The addition of uPA to the HEK-293 cells or ACE-2 overexpressing cells did not affect the expression of apoptotic markers, Caspase-3 and 7, compared to cultured cells that did not undergo treatment (Figure 6G,H). The HEK-293 and ACE-2 overexpressing cells did not show any statistical change in the mRNA expression of autophagy markers, LC3, P62, or Galectin-8 (Figure 6I,J,L). It is worth noting that the LC3 immunoreactive levels decreased when the cultured cells were exposed to the uPA treatment (Figure 6K).

### 3.7. Renal Angiotensin Converting Enzyme 2 (ACE-2) and MasR

While the WT mice showed a significant reduction in renal ACE-2 expression/immunoreactivity following AKI (Figure 7A,D), the αMUPA mice did not display a significant decrease in the expression of this enzyme. This suggests that the ACE-2 may be involved in the protective pathway of the αMUPA kidneys following AKI. However, the expression of renin, a key enzyme of the RAAS pathway which plays an adverse role in hypertension and CKD [42,43], and the expression of MasR, have no statistical change between the WT and αMUPA mice following AKI (Figure 7B,C).

In both the HEK-293 cultured cells and the ACE-2 overexpressing cells, the mRNA expression of the ACE-2 and MasR were not affected by the uPA treatment (Figure 7E,F). However, the ACE-2 protein amount significantly increased following the uPA treatment in both cell types (Figure 7G). As expected, the ACE-2 overexpressing cells exhibit remarkable levels of ACE-2 mRNA and protein immunoreactivity as shown in Figure 7E,G.

### 3.8. Renal Endothelial Nitric Oxide Synthase (eNOS)

The WT mice, susceptible to ischemic insult, demonstrate preserved eNOS immunoreactivity and expression 24 h following AKI (Figure 8A,C). By contrast, the αMUPA mice exhibit a decrease in eNOS immunoreactive levels following AKI (Figure 8A). Importantly, the αMUPA mice demonstrate a decrease in mRNA expression of eNOS both under normal conditions and after AKI (Figure 8C). This is consistent with the fact that the αMUPA mice display higher levels of renal p-eNOS protein amount, the protein that activate eNOS, following AKI compared with WT animals that were subject to AKI (Figure 8B). 

Cultured cells treated with uPA showed a decrease in eNOS immunoreactivity in both the HEK-293 and ACE-2 overexpressing cells. Additionally, both cell types displayed an increase in eNOS mRNA levels (Figure 8D,F). The protein levels of p-eNOS were significantly increased after the uPA treatment in both the HEK-293 and ACE-2 overexpressing cells (Figure 8F). These results suggest that the uPA plays a protective role in αMUPA mouse kidneys against ischemic injury through the activation of eNOS.

## 4. Discussion

One of the primary causes of the high death rate post-AKI is renal ischemia-reperfusion injury (IRI). IRI is a multifaceted process involving various underlying pathways of cell dysfunction and injury. Due to the multifactorial nature of AKI, there is no proven pharmaceutical treatment that can halt its progression or reverse the kidney damage once AKI has occurred. Furthermore, the mechanisms underlying IRI-induced AKI are not yet fully understood. To investigate which pathways are activated or inhibited during AKI, we applied an experimental model of I/R-induced AKI in αMUPA transgenic mice. These αMUPA mice overexpress the uPA gene and are characterized by having increased longevity [44,45]. Therefore, the αMUPA phenotype is similar to dietary-restricted experimental animal models, which are also known to have an increased lifespan.

Previous studies showed that these transgenic mice poses cardioprotection against ischemic and sepsis-induced myocardial infarction (MI) by attenuating inflammatory processes and apoptosis [18,19,20]. Moreover, αMUPA transgenic mice exhibit renoprotection against ischemic AKI, as evidenced by lower post-AKI SCr and BUN levels compared to WT mice; additionally, they display decreased renal levels of inflammatory, apoptotic, and autophagy markers compared to their WT counterparts [21,22]. To investigate which of the pathways play significant roles in nephroprotection against AKI in αMUPA mice, and the involvement of uPA overexpression in this phenomenon, we applied both an in vivo model of AKI in these mice and an in vitro platform where the impact of the uPA treatment was examined in the HEK-293 and ACE-2 overexpressing cells. The present study demonstrates that αMUPA mice exhibit attenuated renal injury 24 h post-AKI, as indicated by a comparatively milder increase in SCr and BUN levels relative to WT mice. Renal mRNA expression levels of NGAL, a sensitive biomarker of kidney injury, were also lower in these mice. Additionally, the αMUPA mice exhibited moderate renal histological alterations in response to I/R AKI, in contrast to the prominent deleterious kidney damage observed in the WT subgroup subjected to the same procedure. Specifically, the αMUPA mice showed less severe renal histological changes following AKI, as evidenced by a reduction in tubular collapse, minimal loss of the brush border, and decreased cellular detachment from tubular basement membranes and necrosis. In line with the in vivo findings, the in vitro uPA-treated HEK-293 cells also showed reduced NGAL mRNA expression. Moreover, the ACE-2 overexpressing cells displayed lower basal expression of NGAL compared to the HEK-293 cells, but did not show further reduction in NGAL following the uPA treatment. These data demonstrate that the uPA and ACE-2 exert an inhibitory effect on the expression of NGAL. The expression of KIM-1, another kidney injury molecule, was comparably increased in both the WT and αMUPA mice following AKI. Similarly, in vitro protocols showed that KIM-1 expression was not affected by the uPA in either the HEK-293 or ACE-2 overexpressing cells. The αMUPA mice showed preserved uPA expression and abundance following AKI compared to the sham-operated mice. Notably, renal uPA receptor expression and immunoreactivity were significantly lower in the αMUPA mice. Interaction of the uPA with its receptor, uPAR, results in innate immune cell activation, cell migration, and inflammation [46,47,48,49]. In this context, acute organ dysfunction, particularly AKI, is strongly associated with elevated blood soluble uPAR levels, especially in critically ill patients with inflammatory diseases [50,51,52,53,54]. In line with these data, the uPAR serves as a pathophysiological mediator of organ damage resulting from systemic inflammatory states. In cultured HEK-293 cells, the addition of uPA led to increased expression and abundance of uPA, while uPAR protein levels significantly decreased following the uPA treatment in both the HEK-293 and ACE-2 overexpressing cells. The endothelium preferentially expresses PAI-1 when angiotensin II levels are elevated [55]. Additionally, when the ACE-2 increases, it converts angiotensin II to angiotensin 1-7, resulting in a decrease in the amount of angiotensin II. Consequently, the αMUPA mice that demonstrated an exaggerated ACE-2 expression/abundance exhibited significantly decreased levels of PAI-1 following AKI compared to their WT counterparts. Consistent findings were observed in cultured cells, where PAI-1 levels notably decreased upon treatment with uPA in both the HEK-293 and ACE-2 overexpressing cells. Notably, the ACE-2 overexpressing cells displayed low basal levels of PAI-1, which were further reduced by the uPA treatment. These results suggest that uPA modulates the uPAR and PAI-1 levels through the ACE-2.

In addition to ischemic damage, the pathophysiological pathways of AKI also involve oxidative stress, mitochondrial damage, inflammatory cell migration, RAAS pathway activation, and decreased NO levels [56]. Therefore, the inhibition of the RAAS pathway (through ACE-2 activation) or the upregulation of eNOS to increase NO production may contribute to AKI attenuation.

PGC1α is a master regulator of mitochondrial biogenesis and function, including detoxification from reactive oxygen species and oxidative phosphorylation [57]. It is another protective mediator that was found to be elevated in the αMUPA mice following AKI (Figure 4). Moreover, it increased significantly following the addition of uPA to the HEK-293 cells. The kidney ranks second only to the heart in terms of mitochondrial abundance among organs with the highest energy consumption [58]. Various types of AKI manifest symptoms associated with mitochondrial damage, prompting research into the role of the PGC1α and mitochondrial biogenesis in aiding kidney recovery from injury. Numerous studies have highlighted the critical role of the PGC1α in mitigating AKI, with growing evidence suggesting that PGC1α loss is linked to the development of renal fibrosis and chronic kidney disease (CKD) [59,60]. Collectively, the PGC1α emerges as another important mediator that aids in maintaining the function of the kidneys in αMUPA mice.

Oxidative stress, excessive stimulation of inflammation, and apoptosis are closely linked to renal I/R AKI [61]. Pro-inflammatory markers like IL-6, TLR4, STAT3, p-STAT3, Cathepsin L, IKB and MAPK were all significantly low in the renal tissue of αMUPA mice. In vitro treatment with uPA significantly decreased the abundance of all pro-inflammatory mediators mentioned above in both the HEK-293 and ACE-2 overexpressing cells. The fibrosis process was inhibited in αMUPA mice following AKI compared to the WT mice, mediated by a reduction in the pro-fibrotic marker TGFβ. Similar results were observed by adding the uPA to both the HEK-293 and ACE-2 overexpressing cells. These findings suggest that the uPA may contribute to the attenuated inflammatory and fibrotic processes observed in response to AKI in αMUPA mice.

Evaluation of mRNA expression and the abundance of apoptotic and autophagy markers revealed reduced basal levels of these markers in the kidneys of the αMUPA mice. The uPA-treated cells did not exhibit any change in fibrosis and autophagy mRNA biomarker expression. Surprisingly, the amount of the autophagy marker LC3 protein decreased upon addition of uPA to both the HEK-293 and ACE-2 overexpressing cells.

It is noteworthy that renal ACE-2 expression and immunoreactivity were both elevated in the αMUPA mice following AKI compared with their WT counterparts. In vitro results demonstrated that direct treatment with uPA increased the mRNA expression of ACE-2 in the HEK-293 cells and significantly elevated the immunoreactive levels of this enzyme in both cell types. These findings suggest that the uPA may protect the kidneys from AKI damage by increasing ACE-2 expression, as observed in the αMUPA mice.

Interestingly, our results demonstrate that activated eNOS (p-eNOS) was significantly higher in the αMUPA mice following AKI compared to the WT mice. Consistent with this finding, direct treatment with uPA decreased eNOS and increased p-eNOS levels in both the HEK-293 and ACE-2 overexpressing cells. Given the beneficial role of NO in AKI [62,63,64], eNOS emerges as another mediator upregulated in the kidneys of the αMUPA mice, potentially contributing to protection against AKI.

## 5. Conclusions

αMUPA transgenic mice demonstrate attenuated damage following AKI compared to WT animals. These mice exhibit preserved renal uPA, which protects the kidney from AKI damage by upregulating the ACE-2, p-eNOS, and PGC1α. These mechanisms lead to vasodilation, protection against oxidative stress, preservation of renal tubular function, and maintenance of renal mitochondria. Additionally, uPA downregulates the inflammatory process, apoptosis, autophagy, and fibrosis, thereby preventing AKI progression and mitigating renal damage after injury. Direct evidence supporting the involvement of uPA in αMUPA nephroprotection was obtained from in vitro experiments, where treatment of the HEK-293 and ACE-2 overexpressing kidney cultured cells with uPA induced upregulation of the ACE-2, p-eNOS, and PGC1α, as well as downregulation of inflammatory, autophagy, and fibrosis markers (Figure 9).

## 6. In Summary

Our core assumption that the mechanisms underlying uPA nephroprotection involve eNOS and the ACE-2 was validated in the current study. We demonstrated that the addition of uPA to the HEK-293 cell culture positively affected the expression of the protective genes mentioned above and negatively impacted the gene expression of pro-inflammatory, fibrotic, and autophagy markers, mirroring the pattern observed in the kidneys of αMUPA mice subjected to I/R-AKI.

Despite the encouraging results, the current study suffers from some limitations: 1—We applied one model of AKI, namely I/R-induced AKI, therefore, our conclusions and interpretations are valid to this etiology; whereas additional studies are required in order to explore whether similar behavior is present in nephrotoxic or septic AKI. 2—The impact of sex on the severity of AKI for 24 h should be explored. 3—The long-term impact of AKI on renal histological changes deserves in-depth investigation. 4—We utilized only one cell line of renal tubular epithelial cells, namely HEK-293; therefore, the application of additional renal cell lines is appealing.

## Figures and Tables

**Figure 1 biomedicines-12-02323-f001:**
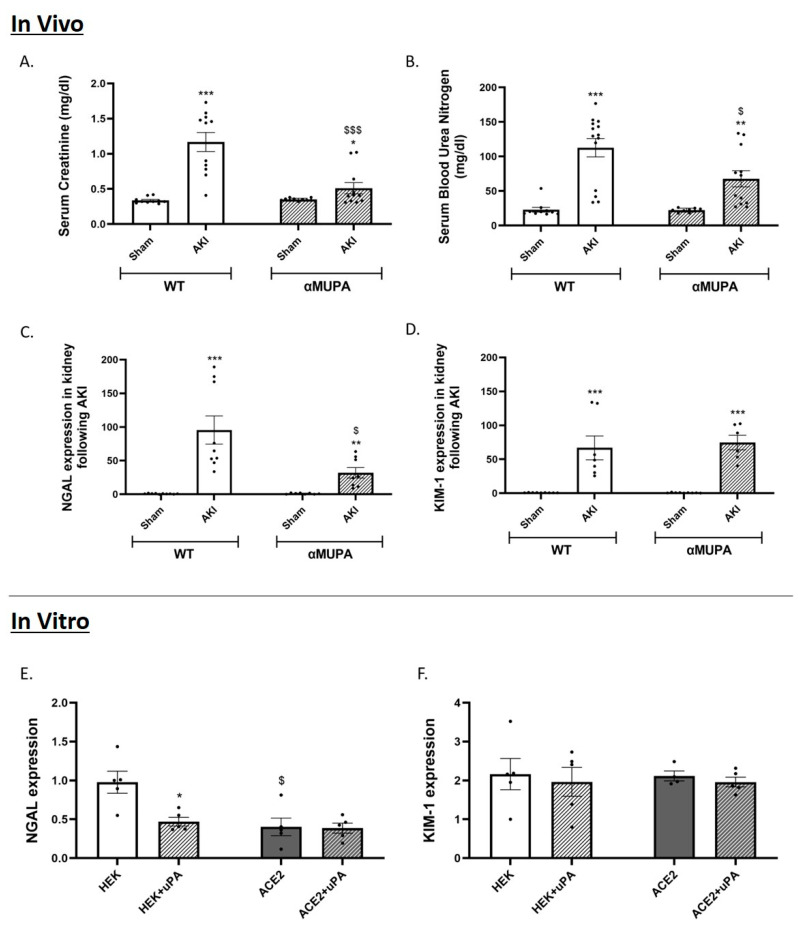
In vivo and in vitro levels/expression of kidney function/renal injury markers in WT and αMUPA mice following sham/AKI (in vivo) and in cultured HEK-293 and ACE-2 overexpressing cells with/out uPA treatment (in vitro): (**A**) Serum creatinine level in mice; (**B**) Blood urea nitrogen (BUN) level in mice; (**C**) q-PCR of renal NGAL in mice; (**D**) q-PCR of renal KIM-1 in mice; (**, *p* < 0.01, ***, *p* < 0.001—Sham vs. AKI in the same mice strain, $, *p* < 0.05, $$$, *p* < 0.001—WT vs. αMUPA which underwent similar procedure); (**E**) q-PCR of NGAL in cells; (**F**) q-PCR of KIM-1 in cells; (*, *p* < 0.05—with vs. without uPA treatment in the same cultured cells group, $, *p* < 0.05—HEK-293 vs. ACE-2 overexpressing cells which underwent similar treatment).

**Figure 2 biomedicines-12-02323-f002:**
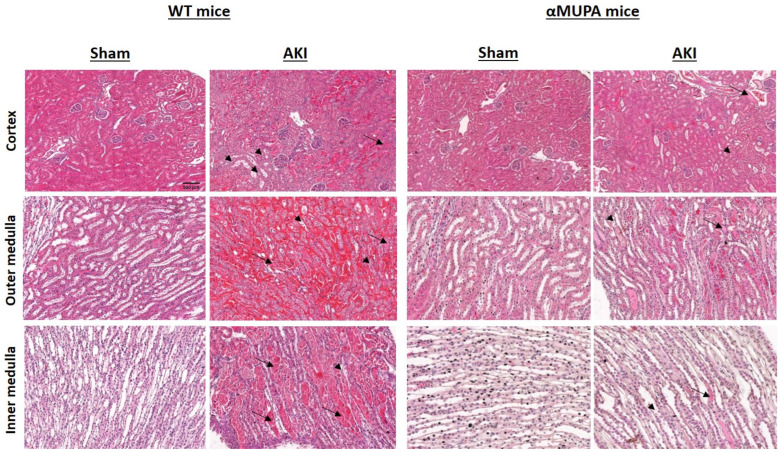
Effects of AKI on renal histology in WT and αMUPA mice: Representative histological images of kidney sections from WT and αMUPA mice, including both sham-operated and AKI models, were taken. The images show cortical, outer medullary, and inner medullary regions stained with hematoxylin and eosin. The first column displays a kidney section from a sham-operated WT mouse, followed by an AKI WT mouse, sham-operated αMUPA mouse, and an AKI αMUPA mouse in the subsequent columns. Long arrows point to areas of tubular collapse, brush border loss, and detachment of cells from the tubular basement membrane, while short arrows indicate areas of congestion. The images were captured at 20× magnification with a scale bar of 100 µm.

**Figure 3 biomedicines-12-02323-f003:**
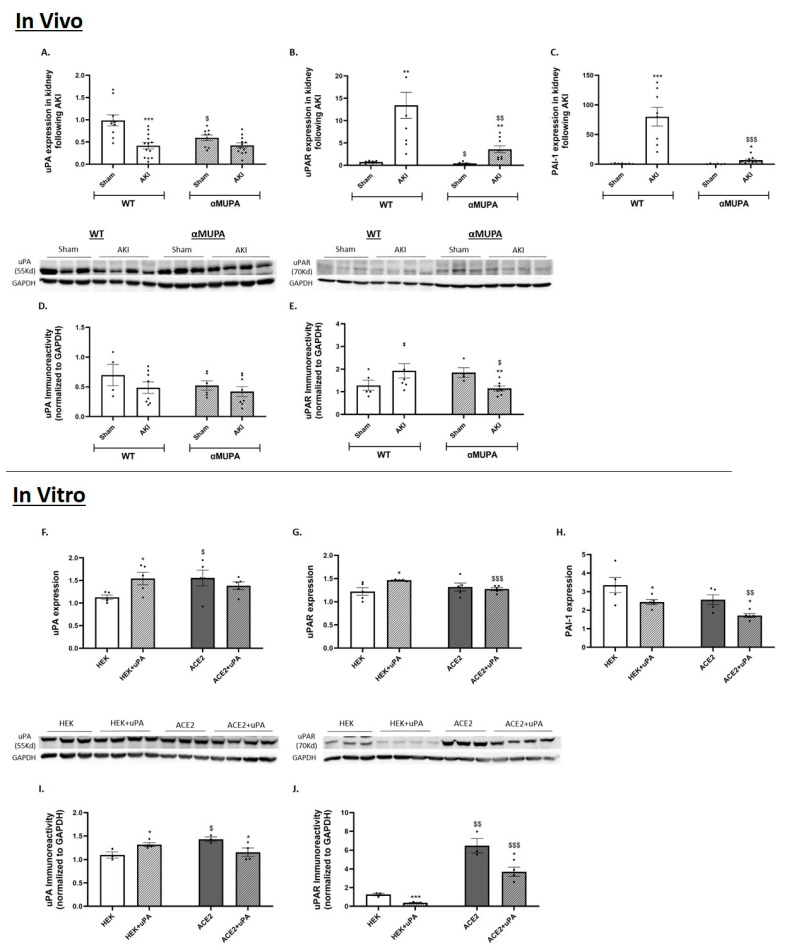
In vivo and in vitro expression/abundance of uPA and uPAR in WT and αMUPA mice following sham/AKI (in vivo) and in cultured HEK-293 and ACE-2 overexpressing cells with/out uPA treatment (in vitro): (**A**) Renal Urokinase Plasminogen activator (uPA) expression in mice; (**B**) Renal Urokinase Plasminogen receptor (PlauR/uPAR) expression in mice; (**C**) Renal Plasminogen activator inhibitor 1 (PAI-1) expression in mice; (**D**) Immunoreactive levels of urokinase plasminogen activator (uPA) in mice; (**E**) Immunoreactive levels of urokinase plasminogen receptor (uPAR) in mice; (**, *p* < 0.01, ***, *p* < 0.001—Sham vs. AKI in the same mice strain, $, *p* < 0.05, $$, *p* < 0.01, $$$, *p* < 0.001—WT vs. αMUPA which underwent similar procedure); (**F**) uPA expression in cells with/out uPA treatment; (**G**) uPAR expression in cells; (**H**) PAI-1 expression in cells; (**I**) uPA abundance in cells; (**J**) uPAR abundance in cells (*, *p* < 0.05—with vs. without uPA treatment in the same cultured cells group, $, *p* < 0.05, $$, *p* < 0.01, $$$, *p* < 0.001—HEK-293 vs. ACE-2 overexpressing cells which underwent similar treatment).

**Figure 4 biomedicines-12-02323-f004:**
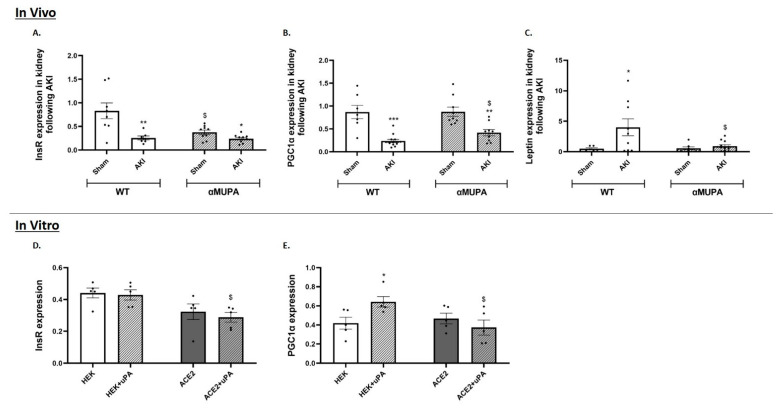
In vivo and in vitro expression of renal leptin, insulin receptor and PGC1-α in WT and αMUPA mice following sham/AKI (in vivo) and in cultured HEK-293 and ACE-2 overexpressing cells with/out uPA treatment (in vitro): (**A**) Insulin receptor (InsR) in mice; (**B**) PGC1α in mice; (**C**) Leptin in mice (*, *p* < 0.05, **, *p* < 0.01, ***, *p* < 0.001—Sham vs. AKI in the same mice strain, $, *p* < 0.05—WT vs. αMUPA which underwent similar procedure); (**D**) InsR in cells; (**E**) PGC1α in cells (*, *p* < 0.05—with vs. without uPA treatment in the same cultured cells group, $, *p* < 0.05—HEK-293 vs. ACE-2 overexpressing cells which underwent similar treatment).

**Figure 5 biomedicines-12-02323-f005:**
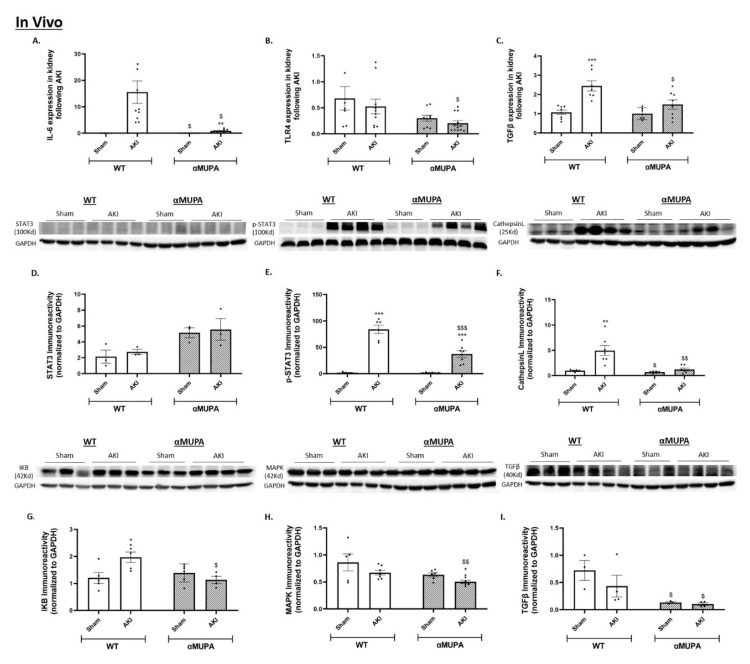
In vivo and in vitro expression/abundance of renal inflammatory and fibrotic markers in WT and αMUPA mice following sham/AKI (in vivo) and in cultured HEK-293 and ACE-2 overexpressing cells with/out uPA treatment (in vitro): (**A**) Expression of IL-6 in mice; (**B**) Expression of Toll like receptor 4 (TLR4) in mice; (**C**) Expression of TGF-β in mice; (**D**) Immunoreactive levels of STAT-3 in mice; (**E**) Immunoreactive levels of p-STAT3 in mice; (**F**) Immunoreactive levels of Cathepsin L in mice; (**G**) Immunoreactive levels of IKB in mice; (**H**) Immunoreactive levels of MAPK in mice; (**I**) Immunoreactive levels of TGF-β in mice (*, *p* < 0.05, **, *p* < 0.01, ***, *p* < 0.001—Sham vs. AKI in the same mice strain, $, *p* < 0.05, $$, *p* < 0.01, $$$, *p* < 0.001—WT vs. αMUPA which underwent similar procedure); (**J**) Expression of IL-6 in cells; (**K**) Expression of TGF-β in cells; (**L**) Immunoreactive levels of STAT-3 in cells; (**M**) Immunoreactive levels of p-STAT-3 in cells; (**N**) Immunoreactive levels of IL-6 in cells; (**O**) Immunoreactive levels of IKB in cells; (**P**) Immunoreactive levels of MAPK in cells; (**Q**) Immunoreactive levels of TGF-β in cells (*, *p* < 0.05, **, *p* < 0.01, ***, *p* < 0.001—with vs. without uPA treatment in the same cultured cells group, $, *p* < 0.05, $$, *p* < 0.01, $$$, *p* < 0.001—HEK-293 vs. ACE-2 overexpressing cells which underwent similar treatment).

**Figure 6 biomedicines-12-02323-f006:**
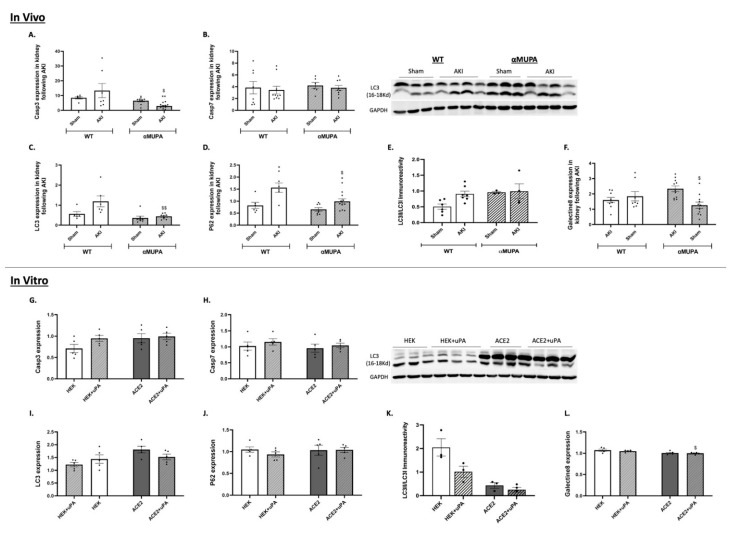
In vivo and in vitro expression/abundance of renal of apoptotic and autophagy markers in WT and αMUPA mice following sham/AKI (in vivo) and in cultured HEK-293 and ACE-2 overexpressing cells with/out uPA treatment (in vitro): (**A**) Expression of Caspase-3 in mice; (**B**) Expression of Caspase-7 in mice; (**C**) Expression of LC3 in mice; (**D**) Expression of P62 in mice; (**E**) Immunoreactive levels of LC3 in mice; (**F**) Expression of Galectin-8 in mice (*, *p* < 0.05, **, *p* < 0.01—Sham vs. AKI in the same mice strain, $, *p* < 0.05, $$, *p* < 0.01—WT vs. αMUPA which underwent similar procedure); (**G**) Expression of Caspase-3 in cells; (**H**) Expression of Caspase-7 in cells; (**I**) Expression of LC3 in cells; (**J**) Expression of P62 in cells; (**K**) Immunoreactive levels of LC3 in cells; (**L**) Expression of Galectin-8 in cells ($, *p* < 0.05—HEK-293 vs. ACE-2 overexpressing cells which underwent similar treatment).

**Figure 7 biomedicines-12-02323-f007:**
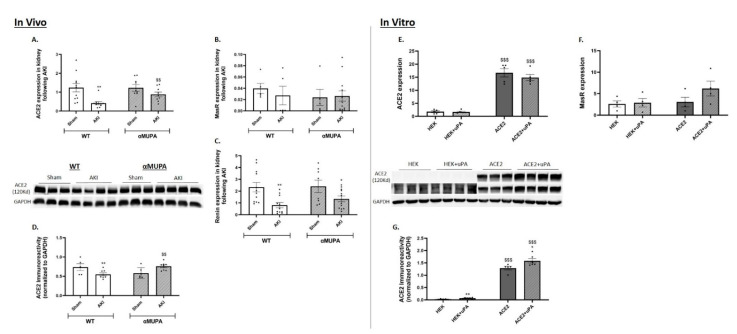
In vivo and in vitro expression/abundance of renal ACE-2, MasR and Renin in WT and αMUPA mice following sham/AKI (in vivo) and in cultured HEK-293 and ACE-2 overexpressing cells with/out uPA treatment (in vitro): (**A**) Expression of ACE-2 in mice; (**B**) Expression of Mas receptor in mice; (**C**) Expression of Renin in mice; (**D**) ACE-2 immunoreactive levels amount in mice (*, *p* < 0.05, **, *p* < 0.01—Sham vs. AKI in the same mice strain, $$, *p* < 0.01—WT vs. αMUPA which underwent similar procedure); (**E**) Expression of ACE-2 in cells; (**F**) Expression of Mas receptor in cells; (**G**) ACE-2 immunoreactive levels amount in cells (*, *p* < 0.05, **, *p* < 0.01—with vs. without uPA treatment in the same cultured cells group, $$$, *p* < 0.001—HEK-293 vs. ACE-2 overexpressing cells which underwent similar treatment).

**Figure 8 biomedicines-12-02323-f008:**
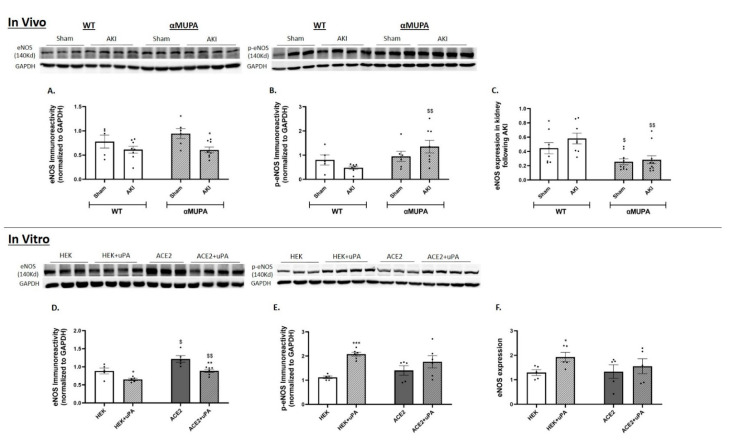
In vivo and in vitro expression/abundance of renal of eNOS and p-eNOS in WT and αMUPA mice following sham/AKI (in vivo) and in cultured HEK-293 and ACE-2 overexpressing cells with/out uPA treatment (in vitro): (**A**) eNOS Immunoreactivity in mice; (**B**) p-eNOS Immunoreactivity in mice; (**C**) Expression of eNOS in mice (*, *p* < 0.05—Sham vs. AKI in the same mice strain, $, *p* < 0.05, $$, *p* < 0.01—WT vs. αMUPA which underwent similar procedure); (**D**) eNOS Immunoreactivity in cells; (**E**) p-eNOS Immunoreactivity in cells; (**F**) Expression of eNOS in cells (*, *p* < 0.05, **, *p* < 0.01, ***, *p* < 0.001—with vs. without uPA treatment in the same cultured cells group, $, *p* < 0.05, $$, *p* < 0.01—HEK-293 vs. ACE-2 overexpressing cells which underwent similar treatment).

**Figure 9 biomedicines-12-02323-f009:**
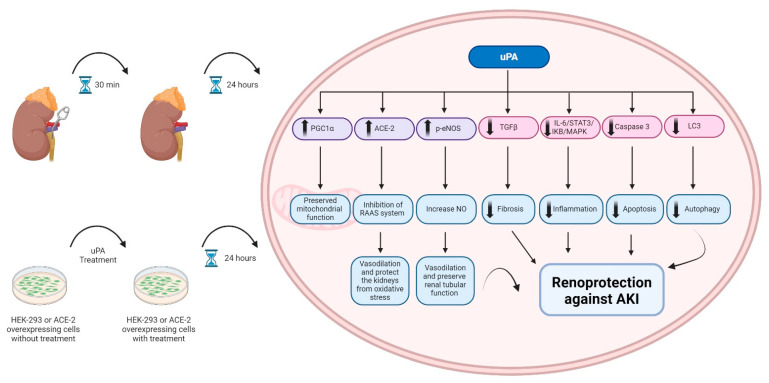
Schematic description of potential mechanisms underlying the nephroprotective actions of uPA against I/R-induced AKI. Figure created with BioRender.com.

## Data Availability

The original contributions presented in the study are included in the article, further inquiries can be directed to the corresponding authors.

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
