# Peer review of "Potential Nephroprotective Effect of uPA against Ischemia/Reperfusion-Induced Acute Kidney Injury in αMUPA Mice and HEK-293 Cells"

_biomedicines, 2024, doi:10.3390/biomedicines12102323_

Round 1
Reviewer 1 Report
Comments and Suggestions for Authors
The manuscript by Alkhaleq et al. reported the potential nephroprotective effect of uPA against Ischemia/Reperfusion acute kidney injury in vivo and in vitro. They concluded that uPA modulates key pathways involved in AKI, offering potential therapeutic targets for mitigating renal damage. I have some concerns below.
1. The nephroprotective action of uPA has already reported by same group in 2023, PMID: 37887341 PMCID: PMC10605904 DOI: 10.3390/cells12202497, The same transgenic mice, the same AKI model, and the same detection indicators, including injury, inflammation, fibrosis, ACE2, apoptosis, and autophagy, etc. The only difference is that the article published in Cells (PMID: 37887341) observed the effects of uPA in female and male mice, while the current manuscript does not specify gender, but at the same time validates the results in vivo on in vitro. Even worse, they did not comment on their previous reports published in Cells (PMID: 37887341) in the introduction or discussion section. Thus, most of the data is repeatedly published, and its novelty is questionable.
2. All results from PCR and WB are displayed irregularly, which should be normalized by internal control (GAPDH) instead of just displaying the ratio of the target protein/gene to internal control.
3. Figure 2, damage should be scored and statistically analyzed. Renal histology should be further evaluated using Masson staining.
4. Regarding the apoptosis, it needs to be validated by using other markers, including WB but not qPCR results of Caspase-3, Bcl2, Bax, and PARP and TUNEL staining.
5. Regarding the autophagy, first, the WB bands of LC3 were not correct, LC3 bands should include two bands, LC3II and LC3I, their ratio preliminarily reflects the level of autophagy. Second, other important markers should be detected, including the protein levels of p62, mTOR, and p-mTOR.
6. Regarding the RAS, why only test ACE2? How about Renin and ACE? In addition, the levels of renin, AngII, and Ang1-7 in the plasma and urine should be detected.
Author Response
We thank reviewer 1 for his/her valuable comments.
Please find bellow itemized response to these comments:
The manuscript by Alkhaleq et al. reported the potential nephroprotective effect of uPA against Ischemia/Reperfusion acute kidney injury in vivo and in vitro. They concluded that uPA modulates key pathways involved in AKI, offering potential therapeutic targets for mitigating renal damage. I have some concerns below:
Comment 1: . The nephroprotective action of uPA has already reported by same group in 2023, PMID: 37887341 PMCID: PMC10605904 DOI: 10.3390/cells12202497, The same transgenic mice, the same AKI model, and the same detection indicators, including injury, inflammation, fibrosis, ACE2, apoptosis, and autophagy, etc. The only difference is that the article published in Cells (PMID: 37887341) observed the effects of uPA in female and male mice, while the current manuscript does not specify gender, but at the same time validates the results in vivo on in vitro. Even worse, they did not comment on their previous reports published in Cells (PMID: 37887341) in the introduction or discussion section. Thus, most of the data is repeatedly published, and its novelty is questionable.
Response: Indeed, the current study covers similar topic of already reported articles: Cells, 2023, PMID: 37887341 PMCID: PMC10605904 DOI: 10.3390/cells12202497, namely the nephroprotective action of uPA. However, the present study differs from these two studies by inducing AKI for shorter period (24h vs. 48h post bilateral renal artery clamping) and the use of in-vitro platform, namely HEK-293 Cells to investigate the nephroprotective effects of uPA, besides the in-vivo model of AKI. The current article cited the already published two articles (Ref. 21, 22) in both the introduction and the discussion. Concerning, the novelty of the current study as compared with the former two articles, the duration of AKI was shorter in order to match the utilization of cell culture, which usually applies few hours up to 24h of exposure to the studied agent. Moreover, the use of cell culture allows examining the direct effects of uPA on renal tubular cells as compared to the physiological impact of uPA genetic manipulation on acute kidney injury in the whole animal.
Comment 2: All results from PCR and WB are displayed irregularly, which should be normalized by internal control (GAPDH) instead of just displaying the ratio of the target protein/gene to internal control.
Response: The PCR results are presented as fold of increase, which is the common way of exhibiting the changes in expression, where we normalized the expression of the studied genes to Rpl13a. We did not normalize the PCR results to GAPDH mRNA since the latter is influenced by I/R, but not Rpl13a mRNA. The WB results were presented as immunoreactive blots as well as their quantification after normalization to GAPDH, where its immunoreactive levels are comparable in the studied samples.
Comment 3: Figure 2, damage should be scored and statistically analyzed. Renal histology should be further evaluated using Masson staining.
Response: Thank you for the comment. As the severity of the renal injury was obvious in the histological slides as evident by cell detachment and congestion, we did not scored the renal damage as it is not very accurate. Concerning, Masson staining, due to the short period of follow up following I/R-AKI (24h), we do not expect to detect fibrotic changes as this process takes many days/weeks to develop.
Comment 4: Regarding the apoptosis, it needs to be validated by using other markers, including WB but not qPCR results of Caspase-3, Bcl2, Bax, and PARP and TUNEL staining.
Response: Thank you again for the note. Unfortunately, we do not have all the required Abs at the moment. Usually, we receive these Abs from overseas companies, and would take a few months to arrive due to the shipping disturbances in the Middle East region.
Comment 5: Regarding the autophagy, first, the WB bands of LC3 were not correct, LC3 bands should include two bands, LC3II and LC3I, their ratio preliminarily reflects the level of autophagy. Second, other important markers should be detected, including the protein levels of p62, mTOR, and p-mTOR.
Response: Thank you for the note. Indeed, two bands of LC3 were obtained in WB analysis. We analyzed the expression of both the active and inactive LC3 normalized to GAPDH in each sample for comparing between the experimental groups. (DOI: 10.1017/S0967199413000269). Currently, our lab does not possess corresponding antibodies p62, mTOR, and p-mTOR, due to difficulties of shipment to this region of the world.
Comment 6: Regarding the RAS, why only test ACE2? How about Renin and ACE? In addition, the levels of renin, AngII, and Ang1-7 in the plasma and urine should be detected.
Response: The measurements of AngII and Ang1-7 in urine is cumbersome and not reliable as these short peptides are unstable and have short t1/2. In addition, the available abs against renin and ACE are not specific and the obtained bands are not obvious/specific.
Reviewer 2 Report
Comments and Suggestions for Authors
The study is novel in its focus on uPA and its protective effects in AKI. The impact could be enhanced by discussing potential therapeutic applications in more detail and by suggesting how this research could inform clinical practice.
- Title could be modefied to "Potential Nephroprotective Effect of uPA Against Ischemia/Reperfusion-Induced Acute Kidney Injury in αMUPA Mice and HEK-293 Cells".
- In methodology, there are a few areas that could be improved:
The sample size and power analysis for the animal experiments should be discussed to justify the chosen number of animals.
More details on the randomization and blinding procedures would strengthen the study's reliability.
The description of the statistical methods is adequate, but including justification for using certain tests over others would provide more clarity.
- Discussion could be improved by:
Reducing repetition: Some points are reiterated multiple times, such as the protective role of uPA and its mechanism of action.
Exploring limitations: While some limitations are acknowledged, there is room for a more critical analysis of potential biases, such as those introduced by using a single transgenic mouse model and the generalizability of in vitro results.
Future directions: The discussion would benefit from a clearer outline of the next steps in research, such as exploring the effects of uPA in other models of kidney injury or investigating long-term outcomes.
- Some sections are overly technical, which may limit accessibility to a broader audience. Simplifying language without losing scientific rigor could improve readability.
Comments on the Quality of English LanguageMinor editing of English language required.
Author Response
We thank reviewer 2 for his/her valuable comments.
Please find bellow itemized response to these comments:
The study is novel in its focus on uPA and its protective effects in AKI. The impact could be enhanced by discussing potential therapeutic applications in more detail and by suggesting how this research could inform clinical practice.
Comment 1: Title could be modified to "Potential Nephroprotective Effect of uPA Against Ischemia/Reperfusion-Induced Acute Kidney Injury in αMUPA Mice and HEK-293 Cells".
Response: We adopted your suggestion.
- In methodology, there are a few areas that could be improved:
Comment 2: The sample size and power analysis for the animal experiments should be discussed to justify the chosen number of animals.
Response: The sample size in each group was determined by professional statistician based on our previous experience with the impact of AKI on serum creatinine (SCr) levels and the standard deviations observed in αMUPA and WT mice.
Comment 3: More details on the randomization and blinding procedures would strengthen the study's reliability.
Response: The study was open one. In contrast to clinical studies, it is uncommon and very difficult to conduct blind studies in animals.
Comment 4: The description of the statistical methods is adequate, but including justification for using certain tests over others would provide more clarity.
Response: Since we have two groups in each mice strain (Sham and AKI), we applied unpaired t-test to compare between the results obtained in AKI vs. sham operation for WT and αMUPA mice.
- Discussion could be improved by:
Comment 5: Reducing repetition: Some points are reiterated multiple times, such as the protective role of uPA and its mechanism of action.
Response: The MS underwent meticulous editing and repetitions were omitted.
Comment 6: Exploring limitations: While some limitations are acknowledged, there is room for a more critical analysis of potential biases, such as those introduced by using a single transgenic mouse model and the generalizability of in vitro results.
Response: We added at the end of the discussion section a paragraph, which refers to the current study limitations.
Comment 7: Future directions: The discussion would benefit from a clearer outline of the next steps in research, such as exploring the effects of uPA in other models of kidney injury or investigating long-term outcomes.
Response: Future directions were added along the study limitations (end of discussion).
Reviewer 3 Report
Comments and Suggestions for Authors
How can the findings from αMUPA mice and HEK-293 cells be translated to human physiology and clinical practice?
What are the long-term effects of uPA treatment on kidney function, beyond the acute phase of injury?
What are the precise molecular mechanisms through which uPA modulates inflammation, apoptosis, and fibrosis pathways in the kidney?
Do compensating mechanisms play a role in the protective effects seen in αMUPA mice, or is uPA overexpression the only cause?
Is it possible to validate the in vitro results with other relevant human kidney cell lines?
How do gender-specific pathways affect the protective mechanisms of uPA in AKI?
Is it possible for uPA to target the kidneys only, without damaging other organs or producing systemic adverse effects?
What role do these results play in the larger scheme of AKI treatment, and how does uPA stack up against other available treatments?
Author Response
We thank reviewer 3 for his/her valuable comments.
Please find bellow itemized response to these comments:
Comment 1: How can the findings from αMUPA mice and HEK-293 cells be translated to human physiology and clinical practice?
Response: It should be emphasized that uPA is expressed in human tissues and present in circulation and its levels are affected by several diseases including AKI and CKD. Therefore, the current study is of physiological and pathophysiological relevance. In this context, αMUPA mice exhibit upregulation of uPA, and uPA is clinically used medication and have been shown to be safe. Additionally, we conducted in vitro on human HEK-293 cells, which represent proximal epithelial cells. We hope that our findings will be incorporated in clinical practice for AKI besides uPA use for thrombosis treatment. By doing so, we hope to improve outcome of certain disease, especially AKI.
Comment 2: What are the long-term effects of uPA treatment on kidney function, beyond the acute phase of injury?
Response: Our studies focused on the impact of uPA on AKI. The present study, we did not conduct long-term experiments concerning the renal effect of uPA in the long run. We hope to expand our research to address this question in future studies.
Comment 3: What are the precise molecular mechanisms through which uPA modulates inflammation, apoptosis, and fibrosis pathways in the kidney?
Response: The molecular mechanism underlying the uPA modulation of inflammation, apoptosis, and fibrosis pathways were elaborated in the discussion, and summarized in figure 9.
Comment 4: Do compensating mechanisms play a role in the protective effects seen in αMUPA mice, or is uPA overexpression the only cause?
Response: Please see figure 9. uPA activates several cellular cascades which mitigates the adverse consequences of AKI, thus improving renal outcome.
Comment 5: Is it possible to validate the in vitro results with other relevant human kidney cell lines?
Response: HEK293 is widely used to validate in vivo studies related to kidney physiology and pathophysiology, as these cells are epithelia ones of proximal tubule properties. Noteworthy, ischemic AKI mainly affects proximal tubule, thus using HEK293 is of relevance to I/R-induced AKI. Moreover, these cells offer advantage of producing fully human post-translational modifications (doi: 10.3390/cells10071667; (doi: 10.3109/07388551.2015.1084266). Furthemore, HEK-293 cells from the proximal tubules are the only human embryonic kidney cell line commercially available for purchase.
Comment 6: How do gender-specific pathways affect the protective mechanisms of uPA in AKI?
Response: Females, which have higher estrogen display substantial resistance to AKI, probably due to higher levels of Phaspho eNOS (p-eNOS). After removal of the ovaries, or L-NAME (eNOS inhibitor) treatment female mice lost their protective capacity against AKI. In both cells and mice, high levels of uPA specifically increased eNOS, which most likely mediates the nephroprotective mechanism. In our previous studies (Cells, 2023, PMID: 37887341 PMCID: PMC10605904 DOI: 10.3390/cells12202497), we have demonstrated that female mice both WT and especially αMUPA exhibited less renal damage than males when were subjected to AKI.
In this study, we did not separate male and female mice, as the HEK-203 cells show no sex differentiation. However, in the presence of uPA, there was a higher p-eNOS immunoreactivity, both in vivo and in vitro.
Comment 7: Is it possible for uPA to target the kidneys only, without damaging other organs or producing systemic adverse effects?
Response: This is a generalized phenomenon. Previous studies showed that αMUPA mice are also cardioprotected, with protection against heart attacks (MI), as well as, sepsis by reducing inflammation and apoptosis. Furthermore, uPA clinical use would not be approved if it causes damage to vital organs.
Comment 8: What role do these results play in the larger scheme of AKI treatment, and how does uPA stack up against other available treatments?
Response: It could be used in clinical AKI, as the drug is safe and is already widely used for other conditions, such as thrombosis.
Reviewer 4 Report
Comments and Suggestions for Authors
The study investigates the nephroprotective effects of uPA against I/R-induced AKI using both in vivo and in vitro models. The study is very well conducted and presented. The use of relevant biomarkers (e.g., NGAL, KIM-1, ACE-2, p-eNOS, and PGC1α) is appropriate for assessment of AKI and kidney protection. In my view, the manuscript quality is high, with only a few minor comments that should be addressed by the authors:
1) The introduction should include a definition or explanation of WT FVB/N and αMUPA mice for non-specialist readers who may not be familiar with these models.
2) In the statistical subsection please specify the software used for the analysis and clarify why all data are presented as mean ± SEM, especially since the figures suggest that not all data were normally distributed.
3) Add the exact p-values in the figures instead of using multiple symbols in the legends. It is also important to define all the abbreviations used in the figures.
4) Although the discussion adequately addresses the study's main findings and their implications, it would be beneficial to compare the results with those of other similar studies.
5) Add the study limitations.
Comments on the Quality of English LanguageThere are some minor typographical errors.
Author Response
We thank reviewer 4 for his/her valuable comments.
Please find bellow itemized response to these comments:
The study investigates the nephroprotective effects of uPA against I/R-induced AKI using both in vivo and in vitro models. The study is very well conducted and presented. The use of relevant biomarkers (e.g., NGAL, KIM-1, ACE-2, p-eNOS, and PGC1α) is appropriate for assessment of AKI and kidney protection. In my view, the manuscript quality is high, with only a few minor comments that should be addressed by the authors:
Comment 1: The introduction should include a definition or explanation of WT FVB/N and αMUPA mice for non-specialist readers who may not be familiar with these models.
Response: αMUPA transgenic mice were generated as previously described by Miskin et al, (Eur J Biochem, 190 (1) (1990), pp. 31-38).
Briefly, DNAs encoding the murine urokinase-type plasminogen activator (uPA) were fused downstream from the promoter-enhancer element of the murine gene encoding αA-crystallin, a protein found exclusively in the ocular lens. Parental WT mice (the NIH inbred mouse line FVB/N (Proc Natl Acad Sci USA, 88 (6) (1991), pp. 2065-2069), served as WT animal.
Comment 2: In the statistical subsection please specify the software used for the analysis and clarify why all data are presented as mean ± SEM, especially since the figures suggest that not all data were normally distributed.
Response: Since we have two groups in each mice strain (Sham and AKI), we applied unpaired t-test to compare between the results obtained in AKI vs. sham operation for WT and αMUPA mice. We applied Prism 10 for statistical analysis.
Comment 3: Add the exact p-values in the figures instead of using multiple symbols in the legends. It is also important to define all the abbreviations used in the figures.
Response: Since the figures are already crowded and expressed as dot blot (required by the journal), added the exact p value which include 5 digits will be difficult and may spillover the relevant bars. *, P<0.05, **,P<0.01, ***, P<0.001 is used widely in the literature.
Comment 4: Although the discussion adequately addresses the study's main findings and their implications, it would be beneficial to compare the results with those of other similar studies.
Response: Unfortunately, no similar studies were done by others. There are only two similar studies were done by us (See ref 21 and 22) and we referred to them in the introduction and discussion. It should be emphasized that αMUPA mice are available to few groups in Israel and among them; we are the only one who works on kidney.
Comment 5: Add the study limitations.
Response: Added- See last paragraph in the discussion.
Round 2
Reviewer 1 Report
Comments and Suggestions for Authors
Most of my comments have not been well addressed. More importantly, this manuscript can be considered as a repetition of the results from the author's previous publications (PMID: 37887341, 38542516), using different presentation methods without providing additional new results and lacking necessary novelty.
Reviewer 2 Report
Comments and Suggestions for Authors
Accept
Comments on the Quality of English LanguageMinor editing of English language required.
Reviewer 3 Report
Comments and Suggestions for Authors
The paper can be accepted in its present form.